# Beta–Gamma Phase-Amplitude Coupling as a Non-Invasive Biomarker for Parkinson’s Disease: Insights from Electroencephalography Studies

**DOI:** 10.3390/life14030391

**Published:** 2024-03-15

**Authors:** Tisa Hodnik, Stiven Roytman, Nico I. Bohnen, Uros Marusic

**Affiliations:** 1Department of Psychology, Faculty of Mathematics, Natural Sciences and Information Technologies, University of Primorska, 6000 Koper, Slovenia; 89211167@student.upr.si; 2Institute for Kinesiology Research, Science and Research Centre Koper, 6000 Koper, Slovenia; 3Functional Neuroimaging, Cognitive and Mobility Laboratory, Department of Radiology, University of Michigan, 24 Frank Lloyd Wright Drive, Domino’s Farms, Lobby B-1000, Ann Arbor, MI 48106, USA; stivenr@med.umich.edu (S.R.); nbohnen@umich.edu (N.I.B.); 4. Department of Neurology, University of Michigan, Ann Arbor, MI 48109, USA; 5Udall Center of Excellence in Parkinson’s Disease Research, University of Michigan, Ann Arbor, MI 48109, USA; 6Ann Arbor VA Healthcare Center, Neurology Service, Ann Arbor, MI 48109, USA; 7Department of Health Sciences, Alma Mater Europaea—ECM, 2000 Maribor, Slovenia

**Keywords:** Parkinson’s disease, electroencephalography, phase-amplitude coupling

## Abstract

Phase-amplitude coupling (PAC) describes the interaction of two separate frequencies in which the lower frequency phase acts as a carrier frequency of the higher frequency amplitude. It is a means of carrying integrated streams of information between micro- and macroscale systems in the brain, allowing for coordinated activity of separate brain regions. A beta–gamma PAC increase over the sensorimotor cortex has been observed consistently in people with Parkinson’s disease (PD). Its cause is attributed to neural entrainment in the basal ganglia, caused by pathological degeneration characteristic of PD. Disruptions in this phenomenon in PD patients have been observed in the resting state as well as during movement recordings and have reliably distinguished patients from healthy participants. The changes can be detected non-invasively with the electroencephalogram (EEG). They correspond to the severity of the motor symptoms and the medication status of people with PD. Furthermore, a medication-induced decrease in PAC in PD correlates with the alleviation of motor symptoms measured by assessment scales. A beta–gamma PAC increase has, therefore, been explored as a possible means of quantifying motor pathology in PD. The application of this parameter to closed-loop deep brain stimulation could serve as a self-adaptation measure of such treatment, responding to fluctuations of motor symptom severity in PD. Furthermore, phase-dependent stimulation provides a new precise method for modulating PAC increases in the cortex. This review offers a comprehensive synthesis of the current EEG-based evidence on PAC fluctuations in PD, explores the potential practical utility of this biomarker, and provides recommendations for future research.

## 1. Introduction

Parkinson’s disease is a neurodegenerative disease characterized mainly by neurodegenerative processes in the basal ganglia (BG) region [1]. Structural changes due to neurodegeneration lead to disturbances in various brain networks, which translate into functional changes in the brain. As a result, motor symptoms such as tremor, bradykinesia, rigidity, dystonia, and freezing of gait (FOG) occur as the disease progresses [2]. Although considerable efforts have been made to uncover the underlying causes of neurodegenerative processes in PD, their origin remains elusive and an active area of research [3].

Pathological, structural, and functional changes in PD are reflected in changes in the electrophysiological circuitry of brain networks [4]. The use of electroencephalography (EEG), magnetoencephalography (MEG), and functional near-infrared spectroscopy (fNIRS) to identify functional and electrophysiological changes caused by neurodegenerative processes has contributed significantly to our understanding of pathological brain conditions. These methods are non-invasive and their application is, therefore, not limited to patients undergoing surgery [5,6,7]. Electroencephalography (EEG) offers numerous advantages, particularly in the clinical field, as it is easily accessible, cost efficient, and time efficient. It is characterized by the recording of neuronal activity with superior temporal resolution, albeit with a compromise in spatial resolution. The ability of EEG to monitor electrophysiological fluctuations over extended periods of time is demonstrated by its effective use in longitudinal studies [8] and applications in challenging/extreme environments [9]. Recent advances in mitigating motion-induced artifacts have increased the utility of EEG in mobile settings, thereby bolstering the ecological validity of the measurements obtained [10]. As a well-established and thoroughly investigated method, EEG enables the integration and comparison of new results with existing data, increasing the reliability and validity of new research findings [11].

Several electrophysiological changes have been associated with PD [12]. An increase in phase-amplitude coupling (PAC) between the beta and gamma bands detected over the motor cortex has been associated with motor abnormalities of PD [13,14]. PAC is a form of cross-frequency coupling (Figure 1), a crucial mechanism for information processing in the brain. It is involved in many higher-order cognitive processes that contribute to the successful accomplishment of everyday tasks [4]. While the role of PAC in cognitive tasks is well investigated, the role of PAC in movement and the modulation of PAC by movement is less known [15]. Beta–gamma PAC reliably distinguishes PD patients from healthy controls and reflects medication status [16,17]. The increase was found both in measurements at rest and in measurements during hand movements [17], while in ECoG measurements, the increase was also found during walking. This increase correlated with the occurrence of FOG episodes and could be mitigated by DBS [18]. Biomarkers that respond to disease states could be used in the future to more accurately customize and modulate treatment based on variations in Parkinson’s motor symptoms [14,19]. However, the association with FOG episodes in PD patients and their alleviation using DBS [18] warrants further research to identify possible correlations with specific disease subtypes. These results provide a reliable basis on which the parameters of an accurate and sensitive biomarker could be determined [20]. This review of the existing literature addressing EEG-detectable PAC changes due to PD explores the characteristics and potential utility of a non-invasively detectable biomarker of motor status.

### 1.1. The Crucial Role of Synchronization in Brain Function

Neural communication in the brain is facilitated by electrical impulses resulting from ion movement during synaptic transmission. The electric signal recorded by EEG is generated by the ionic currents in the postsynaptic dendritic membrane of the pyramidal neurons in cortical layers IV-V [21]. Insights from electrophysiological research have significantly advanced our understanding of brain function [22]. Different frequency bands detected by EEG, delta (1–4 Hz), theta (4–8 Hz), alpha (8–12 Hz), beta (12–30 Hz), and gamma (>30 Hz) each possess unique spatial and temporal characteristics [22], which are associated with different levels of alertness, consciousness, and cognitive processes [19]. Bands oscillating at lower frequencies are thought to be responsible for synchronization across larger spatial scales, while higher frequencies are implicated in the modulation of computational processes within localized neural circuits [23]. Different signal types tend to oscillate in similar frequencies even though they might have a different locus of origin [22]. Oscillations occurring in the frequency of the beta band (12–30 Hz) are typically observed within the motor system-related cortical and subcortical structures [24], while high gamma amplitude is thought to carry information related to motor tasks such as movement types and onset timing [25]. Gamma oscillations are also viewed as a reflection of the spiking of neuronal populations and its increase is a surrogate for local neuron activity [16]. The ability of the cortex to generate gamma oscillations is dependent on the state of consciousness and arousal [26].

Coordinated interactions between bands provide a mechanism for spatial and temporal integration of information across cortical and subcortical structures, therefore regulating information integration of dispersed neural populations, allowing them to operate in parallel [27]. A substantial amount of brain activity is required for successful everyday functioning and is dependent on the process of synchronous neural firing [22,28]. Synchronization of neural activity can be observed on the microscopic level between local neural populations or span across multiple cortical and subcortical regions, even across hemispheres, and can occur within one frequency band or between separate bands [4]. It is thought that synchronization within the beta band presents a major aspect of basal ganglia (BG) activity, a core part of the motor system [29], while gamma-band synchronization is a fundamental operation mode of activated cortical networks [28]. It has been determined that movement initiation and execution are mediated by synchronization processes mainly in the beta band, with synchronization facilitating movement inhibition and desynchronization (synchronization suppression) occurring during movement initiation [29,30,31,32]. During movement tasks, beta synchronization and its suppression have been related to movement types, with the duration of suppression relating to the complexity of the movement, and the latency of suppression predicting the onset of movement. The desynchronization in beta is observed to give way to a spike in gamma oscillations during movement execution [4].

While synchronization in one band is thought to reflect local computation, the integration of activity from different brain regions is achieved through the synchronization of different frequency bands, termed cross-frequency coupling (CFC) [33]. In the human cortex, CFC can be found in the motor cortex [18,20], over the motor cortex [34], in the basal ganglia (BG) region [35], and in the prefrontal cortex, as well as the interaction between these regions [36]. CFC can be observed as phase-to-phase coupling, phase-to-frequency coupling, phase-to-amplitude coupling, and amplitude-to-amplitude coupling [37]. Phase-to-amplitude coupling (PAC) is a form of CFC, in which a phase of a low-frequency rhythm (m) modulates the amplitude of a higher-frequency oscillation (n) [38]. In amplitude coupling, a slower frequency dictates a temporal code and acts as a carrier frequency, while phase coupling improves temporal precision and is dictated by the spiking activity of the higher frequency [39]. This results in temporally coordinated neural excitability of dispersed neural populations, which in turn have an increased synaptic activity and elevated impact on their select target network [40].

There are many explanations for the occurrence of non-pathological phase-amplitude coupling. One functional explanation attributes the occurrence of PAC to the synchronized firing of underlying neuronal populations oscillating at different frequencies, which can be observed in the hippocampus [41]. Since PAC is an occurrence between a higher and lower frequency oscillation, another common interpretation attributes PAC to the locking of the spiking of neurons generating the higher frequency, which is carried out by locking its activity to the low-frequency oscillation [42]. PAC is spatially distributed and phase diverse [27], which can be observed as a substantial variability in PAC-coupled frequencies and their spatial distribution [35]. This is as a result of the way different neurons produce oscillations and the target network they are trying to reach [39]. Variation in frequency and phase provides flexibility in the selective routing of information, which is crucial for providing communication between distributed networks [27].

This supports the crucial role CFC has in linking relevant brain regions, which results in sensory information integration and neural computation required in tasks such as attention, learning, working memory, and memory consolidation [20,43,44]. PAC is a means of sequence encoding, storage, retrieval of information, and communication between local networks, as well as macroscale systems in the brain [45]. It can increase neural information flow and improve the behavioral performance of actions controlled by distant neural populations [40]. Abnormalities in PAC have been connected to disruptions in higher-order cognitive processes such as alpha–gamma PAC decrease, which was detected in Alzheimer’s dementia patients in comparison to healthy controls [46], while topographical changes in delta–theta PAC were observed in PD patients with mild cognitive impairment during a visual oddball task [47]. Apart from its crucial role in higher-order cognitive processes [20], a significant impact of PAC can be attributed to coordinating neural populations responsible for movement execution [15]. In an ECoG study, a movement-related reduction in healthy human subjects’ and PD patients’ PAC has been reported in the sensorimotor cortex, where the strength of PAC during waiting correlated with movement selectivity during execution, suggesting the role of PAC in motor representation [25]. In a primate Parkinsonian model, PAC of low frequency (4–10 Hz) and high frequency (gamma, 50–150 Hz) has been shown to correlate with motor symptom progression and reached significance in the final stages of symptom development [48]. Furthermore, the primate model reveals PAC strength increases during sleep; however, it is not indicative of Parkinsonism in the unconscious state [48]. Studies show that PAC can be modulated by movement and responds to different movement types [13,17]. Changes in PAC during movement have also been applied to differentiate PD patients from healthy participants, which has been confirmed in studies investigating PAC modulation during unilateral hand movements [13,49] or during walking [18].

### 1.2. What about Nonsinusoidal Brain Oscillations?

Generally, oscillations in the brain are proposed to hold a typical sinusoidal shape, which is a notion underlying all spectral analysis methods. Conversely, many typical nonsinusoidal shapes have been recorded in the brain. A closer investigation of waveform shapes has revealed their active role in neural communication and has been proposed as a separate means of carrying information. Changes in waveform shapes have been observed consistently across recordings, indicating that the change of shape is specific to a task or brain region and is not a consequence of the method used [33]. Nonsinusoidal shapes also change the properties of oscillations, as they have been shown to result in increased synchronization and respond differently to DBS [40]. A typical example of such an oscillation is the mu rhythm, an oscillation in the bandwidth of 8–12 Hz, resembling the Greek letter μ [33].

Beta oscillations have been observed to have a nonsinusoidal shape, resembling a sawtooth. The shape follows a pattern of a rise, decline, another rapid rise, and then a sharper decline, forming a double peak oscillation [33]. Recent research has observed the change of nonsinusoidal oscillations as potentially being characteristic of PD pathology, evident as an increase in the peak and sharpness ratio. This change was observed to be medication-responsive [50]. Both the increase in steepness ratio, reflecting the steepest voltage change in one sample between each peak and trough, as well as the increase in sharpness ratio between average peak and trough sharpness, correlated with PAC is observed in PD patients [50]. It has been hypothesized that the detection of PAC deemed statistically significant can reflect one oscillator with sharp edges, further proving the need to improve techniques suitable for detecting asymmetrical waveform shapes [40]. A possible interpretation ascribes what some studies interpreted as an increase in beta–gamma phase-amplitude coupling, to the nonsinusoidal shape of the lower frequency oscillation, which creates a higher frequency component with a consistent phase [51]. The exploration of nonsinusoidal oscillations in the brain challenges traditional views of brain oscillations and opens new avenues for the development of diagnostic tools and therapeutic interventions, particularly in the context of movement disorders such as PD [40,50]. As we continue to refine our techniques for analyzing waveform shapes, we may uncover further insights into the complexities of brain function and dysfunction, which could represent a significant advancement in our understanding of neural communication and brain dynamics. However, research regarding this topic is severely limited, as techniques that can be used to characterize waveform shapes are lacking [51].

## 2. Current Findings Regarding PAC in PD

While PAC is an invaluable means of transporting integrated information, which is crucial for brain function in the fast-changing everyday environment, excessive PAC has been linked to several neurodegenerative and neuropsychiatric disorders [39].

In Parkinson’s disease, an increase in gamma and beta phase-amplitude coupling has been observed. These observations were first made using invasive techniques for detecting changes in brain states [52,53] and have recently been confirmed in non-invasive EEG recordings [17,34]. A pathological increase in beta–gamma PAC has been found in the STN, over the primary motor cortex, and the sensorimotor cortex, as well as in the interaction between these areas [34,54]. Excessive beta–gamma PAC over the motor cortex is thought to trap the underlying neuronal populations in an inflexible pattern of activity, leading to difficulties in movement, initiation, and execution [55]. For example, rigidity and dystonia have been correlated with increases in beta–gamma PAC [55], while the increase in PAC during walking trials has been associated with the occurrence of FOG [18]. In relation to beta synchrony increases, bradykinesia and dystonia have been correlated with pathological beta bursts, transient events of high amplitude beta oscillations [56]. These findings indicate the intricate dynamics underlying PD motor pathology and state that no single mechanism can be attributed to the sum of motor deficits in PD patients, warranting further investigation.

Currently, five studies (for more details, please see Table 1) present findings of EEG-detected beta–gamma PAC in Parkinson’s disease, providing reliable groundwork for exploring the role of PAC in PD pathology [13,16,17,34,50]. Three of these studies present data from resting state recordings [16,34,50], one provides a comparison of resting state and movement recordings [17], and one presents data from recordings made during repetitive movements [13]. One study [17] mentioned controlling patient inclusion criteria based on the presence of tremors (resting tremor score > 1), while two studies [13,34] mentioned a tremor-based exclusion for patients with significant tremors (resting tremor score ≥ 2). In all studies mentioned, tremors were assessed using UPDRS III.

All studies investigating beta–gamma PAC present a statistically significant increase in beta and gamma PAC in PD patients off medications compared to healthy participants, which was localized over the sensorimotor electrodes. One study, focusing on the spatial distribution of PAC, determined six areas of interest in which beta–gamma PAC was significantly increased. The highest increase was observed over the premotor cortex, with regions of interest (ROIs) also composed of the inferior frontal cortex, dorsolateral prefrontal cortex, primary motor cortex, and Brodmann’s area 1, 2, and 3 (primary somatosensory cortex and primary sensorimotor complex), when considering both hemispheres. When considering separate hemispheres, no regions of interest reached statistical significance in the ipsilateral hemisphere to the more affected side of the hemibody. Every ROI but the inferior frontal cortex reached statistical significance on the contralateral side of the affected hemibody [34]. Another study reports a significant increase in PAC on the contralateral but not the ipsilateral hemisphere to the more affected side of the body, while others made no such comparisons [17].

One proposed advantage of PAC has been the ability to detect coupling changes based on the data provided by only two electrodes (C3 and C4), since these electrodes are closest to M1, as presented in three studies [16,17,50]. Two of them presented results from the same dataset [16,50], while one study that performed analyses on a different dataset contradicted this finding and found no statistically relevant increase in other electrodes over the sensorimotor cortex [34]. It remains unclear whether data from only two electrodes is satisfactory to provide statistically relevant increases which differentiate PD patients from healthy controls. No correlation between either beta [17] or gamma power [34] increases and PAC has been found, indicating that the increase in PAC is unlikely to be caused by changes in spectral power.

Every study researching EEG-measured PAC has found the phenomenon to be medication responsive, meaning that the application of medication affected the synchrony increase. This decrease in PAC has been correlated with the medication-induced motor symptom alleviation measured using the UPDRS scale [17,50]. This was shown by making comparisons between both on-medication and off-medication patients, as well as by making comparisons with healthy controls. All studies found differences between off-medication patients and healthy controls, while three studies found differences in both off- and on-medication patients in PAC levels over the sensorimotor cortex [16,17,50]; two studies made no such comparisons [13,34]. One study had to change the reference scheme in order to achieve these results [50], pointing out the sensibility of PAC to the reference scheme used. However, in the same study, changes between on- and off-medication states were detected based on the analysis of the beta frequency waveform shape alone, which was not contingent on the reference scheme used [50].

More sensitive PAC measures combined with longer recording sets have been proposed to be able to differentiate between patients and healthy controls more accurately, as well as to be able to differentiate between patients individually, as only group comparisons have been made as of yet [16]. Studies excluded patients based on tremor scores and all studies excluded the possibility of PAC being the result of movement artifacts. However, a direct correlation of PAC to tremor was not discussed. A study recording beta and high-frequency oscillation (HFO, 200–400 Hz) coupling found that beta synchrony in the STN was attenuated during the emergence of tremors [52], while another study found an increase in STN and cortical coherence that was correlated to tremor strengths [54]. Tremors could therefore disrupt the results of PAC EEG recordings, while the relationship between PAC and tremors, therefore, remains an open area for investigation. 

Efforts to improve the biomarker sensitivity have been made by including waveform shape as an inclusion criterion [17,50]. Two studies mentioned using beta waveform shape as a sensitivity measure [17,50]. Both were able to make between-group differentiation in off-medication patients and healthy controls; but in the study of Miller et al. [17], there was no decrease in the sharpness ratio caused by medication, which was observed in the dataset of Jackson et al. [50]. Beta sawtooth-like shape can contribute to increased synchronization in this band and, in PD, the pathological change of the shape can be the underlying mechanism for pathologically increased beta band synchronization and PAC. Using the asymmetrical waveform shape of beta as an inclusion criterion for significant beta–gamma PAC shape has been used as a means of improving the reliability of the biomarker [50], suggesting a further possible correlation. Comparisons on an individual level, based on waveform shape detected non-invasively, have not yet been presented, warranting further research regarding the role of beta waveform shape in PAC, implementing waveform-sensitive techniques.

Two datasets have examined PAC during movement initiation in non-invasive EEG recordings [13,17]. In the study executed by Gong et al. [13], the role of PAC in repetitive movements was investigated, with the goal of determining whether PAC behaves differently depending on the type and velocity of the movement performed. Differences were observed in fast tapping, slow tapping, and the pressing task. PAC was observed to be increased in the pressing task in PD patients compared to controls, while no changes were observed in tapping tasks, suggesting PAC cannot be deemed sufficient for the sum of motor impairment in PD. The correlation of PAC to beta power was present but inconsistent. Interestingly, a movement initiation mechanism of PAC was observed [13]. This mechanism of neural discharge resembled the preparatory neural activity mechanism of motor control [57] according to the dynamic systems theory of motor control [58] and can be described as a decrease, peak, and rebound in PAC strength [13]. This theory centers around the notion that systems generate the firing pattern of activity, which then produces movement driven by initiating forces that move the body in a way that achieves the desired goals [58]. This mechanism is apparent around movement onset and reflects preparatory mechanisms around motor transitions; however, it is not movement-type specific [13]. This was less apparent in PD patients than in healthy controls, as the decrease and rebound of PAC were significantly smaller [13]. Further research into the role of PAC during movement is needed to fully explain the mechanism and role of PAC in movement preparation and initiation.

## 3. Discussion

In PD, the degeneration of neurons in substantia nigra results in the loss of nigrostriatal dopaminergic projections. Dopamine has an integral function in regulating the operation of the striatum and the depletion of this neurotransmitter in the basal ganglia region causes severe disruptions in the BG network (cortico-basal ganglia-thalamo-cortical loop, (CBGTC)) [59]. The loss of dopaminergic inputs to the striatum leads to hypoactivity of pro-kinetic direct pathway striatal projection neurons (bearing excitatory D1 receptors) and hyperactivity of anti-kinetic indirect pathway striatal projection neurons (bearing inhibitory D2 receptors), which is thought to explain the bradykinesia and rigidity symptoms commonly observed in PD [1].

Beta oscillations are thought to be generated in the basal ganglia and there are many models that attempt to explain the intricate changes by which neuronal degeneration in PD leads to an increase in beta synchrony [60]. A more recent animal-based model attributes the increase in synchronization to an increased interaction between two separate oscillators in the BG. These oscillators are the globus pallidus externa, subthalamic nucleus loop (GPe-STN, the STN loop), and the loop of fast-spiking neurons, globus pallidus externa, and inhibitory dopaminergic neurons (FSN-GPe-D2, the STR loop). Through dopamine depletion, the interplay of these loops mediated by GPe is pathologically increased, resulting in two separate oscillators synchronizing into one oscillator, causing the increase in beta band synchronization [60]. This increase in beta band activity has been shown to correlate with motor symptom severity and increase with disease progression [42]. However, beta activity in PD patients can resemble that of healthy subjects, but only in the presence of dopaminergic therapy [61], and has been shown to respond to therapy in a dose-dependent manner [16,17]. Phase amplitude coupling between beta and gamma observed in PD is thought to reflect the increase in neural entrainment in BG caused by the degenerative processes [16,50]. Furthermore, PAC has been shown to attenuate during DBS stimulation of the STN, supporting the notion of its modulation by the functioning of the BG [18,60]. Its responsiveness to dopaminergic modulation [16], as well as its correlation with improvements in rigidity [1], suggest a partial modulation by dopamine. This change was correlated with clinical motor symptom alleviation, further indicating that pathological PAC has a role in producing motor pathology in PD, although it cannot account for the sum of motor deficits in patients [17,18]. This shows that PAC is able to differentiate PD patients and healthy controls, as well as indicate medication states and could, therefore, be used to aid in treatment modulation if the sensitivity of the biomarker is improved. PAC characteristics in resting state over time are not well known and suitable parameters that would provide a representative dataset have not been proposed [50].

There are no reliable findings proving that waveform shape alone could reflect medication states [17,50]. The role of oscillation shape in PD pathology could help explain the underlying mechanisms of increased PAC coupling in PD. If PAC is dependent on beta shape, it would suggest that PAC is a result of one underlying mechanism, while if the shape proves to be unrelated to PAC, it would suggest that beta and gamma are produced by two separate oscillators [35].

While dopamine depletion is the main cause of PD pathology, the loss of this neurotransmitter cannot account for the sum of PD characteristic pathological changes in the brain. Changes in D1 and D2 modulation have been connected to alterations in striatal cholinergic interneuron activity. This modulation affects the GABAergic component of the EPSP and the muscarinic component of IPSP, through the modulation of calcium currents through N-type channels. Through this mechanism, the loss of dopaminergic neurons due to PD results in a lack of inhibition of cholinergic neurons in the striatum [62].

Cholinergic interneuron activation in M1 can also be observed in EEG changes as an increase in beta power in all cortical layers and gamma power in deeper layers [63].

The modulation of striatal cholinergic interneuron (SChI) activation is connected to the elevation of striatal oscillations; however, this effect does not translate to M1, confirming that basal ganglia are the generator of beta oscillations in that region. SChI activation was also connected to changes in locomotion and resulted in pathological changes resembling that of Parkinson’s disease, such as hypokinetic movement, and increased coherence between the cortex and basal ganglia, in addition to the previously mentioned increase in beta oscillations in M1 and the striatum [63]. Given the essential role of M1 in the physiology of walking, research into excessive synchrony in this region could be fundamental to understanding the pathophysiology of FOG in PD [18].

Deep brain stimulation (DBS) of the STN is an effective surgical treatment for PD [64]. DBS has the potential to interrupt the flow of abnormal information within the basal ganglia’s circuits, possibly by separating the incoming and outgoing signals of the STN, which can lead to a more normal pattern of brain activity [18]. The effectiveness of this method tends to diminish over time and the method is associated with various side effects [63]. Furthermore, DBS is administered continuously and adjustments due to disease progression must be made manually [14]. Therefore, neuromodulatory methods for real-time adjustments could improve the efficacy of DBS and reduce unwanted side effects, by administering changes based on motor symptom variation detected by electrophysiological changes [19].

Neuromodulator approaches based on feedback from PAC modulation have been proposed as tools to quantify motor impairment in patients and to aid in therapy modulation by adjusting stimulation parameters [18,19,20,23]. This approach proposes the detection of changes in motor symptoms based on PAC increases, which would then lead to adjustments in stimulation, improving the efficacy of this technique [23]. A phase-targeted stimulation technique has been proposed as a promising method of modulating PAC increases in the motor cortex [19]. This would require a phase-dependent stimulation, delivering specifically timed pulses to the carrier frequency phase selectively modulating network activities [19]. A stimulation locked to the beta peak has been shown to increase PAC, while the application of an opposite-phase stimulation decreased beta–gamma PAC [20]. While phase-targeted stimulation altered pathological processes, it evidently did not alter other physiological processes in the basal ganglia-thalamo-cortical (BGTC) network. As PAC alleviation has been connected to motor symptom improvement [16,17,18], this modulation could potentially lead to symptom improvement with fewer side effects [19]. However, for PAC-based neuromodulation to be viable, it is crucial to establish individual PAC patterns for patients, which will provide the necessary therapeutic windows for a PAC-based PDS neuromodulation system [19]. Furthermore, research regarding sleep-affected PAC changes in Parkinsonian patients and the effect this can have on closed-loop DBS is needed in order to further illuminate the possible interactions [48]. While no studies utilize mobile EEG technology to investigate PAC during walking, an ECoG study reports promising findings. The study investigated beta–gamma PAC in M1 during walking trials in patients diagnosed with freezing of gait (FOG) and treated with deep brain stimulation (DBS) [18]. The highest PAC was observed during standing, which corresponds with previous findings reporting that PAC reduces during movement [13]. In walking trials, PAC was pathologically increased in PD patients compared to healthy participants; however, PD participants showed variability in PAC increases [18]. During stable walking, PAC increases were correlated to freezing severity in patients. Furthermore, PAC increases correlated with FOG occurrences, during which PAC was significantly increased compared to stable walking. PAC attenuation caused by DBS correlated with fewer FOG occurrences; however, FOG was less common in walking trials with DBS, even when PAC was not significantly attenuated, suggesting there are other possible mechanisms responsible for FOG alleviation [18]. PAC did not correspond to walking velocity; therefore, changes in velocity caused by FOG are unlikely to be the underlying cause of PAC changes [18].

In their study, a new model for the occurrence of FOG is proposed, named the ‘bandwidth model’, in which they explain FOG occurrences by correlating them to PAC changes. In the model, baseline occupation represents the occupation of cortical processing resources, which reflects the degree of motor impairment and can be quantified by measuring PAC increases in M1 during trials without FOG occurrences. Dynamic fluctuation reflects the cognitive burden of a current task. If dynamic fluctuation surpasses the bandwidth limit, FOG occurs. The area between baseline occupation and bandwidth limit is available bandwidth, which represents available resources for neural processing [18]. PAC and its attenuation by DBS can, therefore, be used to quantify the amount of motor impairment in PD patients and to quantify symptom alleviation by DBS, which is thought to lower the baseline occupation (PAC during non-freezing trials) and raise the bandwidth limit [18]. 

Phase-amplitude coupling between the beta and gamma bands is a reliable biomarker for Parkinson’s motor function, correlating with bradykinesia, rigidity, and freezing of gait. It successfully distinguishes between Parkinsonian patients and healthy subjects, while correlating to the degree of motor impairment in patients and reflecting medication states, as well as the level of medication-induced symptom alleviation. It can be utilized as a biomarker for improving the efficacy of DBS therapy, while it further provides a new target for stimulation. Phase-targeted DBS can target beta–gamma PAC directly, by locking the stimulus to the beta phase, resulting in PAC modulation.

Future research could expand the existing knowledge on the use of adjustable phase-dependent DBS to determine the clinical applicability of such treatments. Precise parameters of PAC under different conditions, as well as the sensitivity of PAC to motor conditions, should be investigated in future studies to improve the reliability of this marker and enable its application in PD treatment.

## 4. Limitations

This review is conducted with the aim to present a focused phenomenon and does not encompass the sum of literature available which describes the theme. Therefore, the data presented in the review could be biased by the chosen literature, limited to the English language. Furthermore, the phenomena discussed are still an open area of research, warranting further investigation in order to be understood in full.

## Figures and Tables

**Figure 1 life-14-00391-f001:**
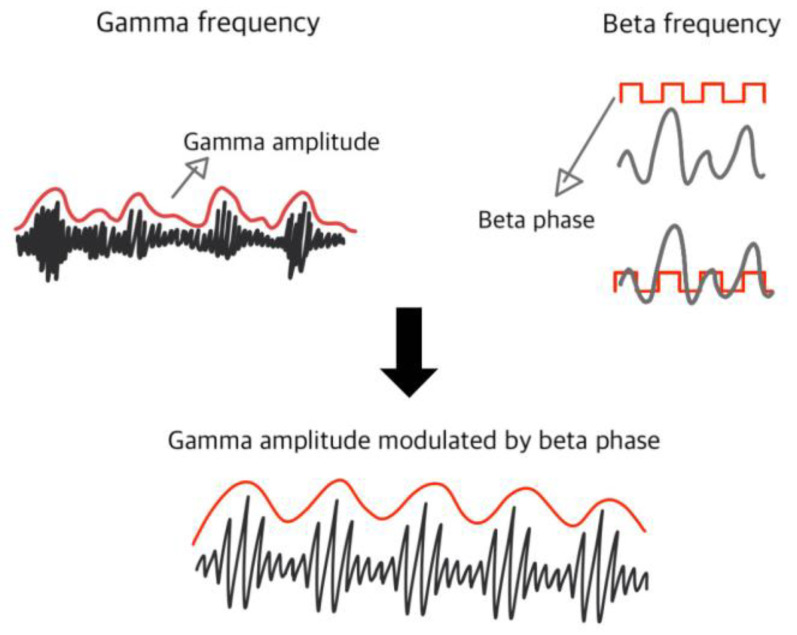
Phase-amplitude coupling (PAC).

**Table 1 life-14-00391-t001:** Summary of findings of EEG-detected beta–gamma PAC in Parkinson’s disease.

Citation	Measurements	Participants	ON vs. OFF	OFF vs. HC	ON vs. HC	Main Finding	Waveform Shape Analysis	Effect Size (PD_OFF vs. HC)
Swann et al. (2015) [16].	32 channel, 3 min RS, EO	14 patients	x	x	x	PAC calculation can be based on only two electrodes (C3/C4).		effect size: 0.83, *p* = 0.009
Jackson et al. (2019) [50].	32 channel, 3 min, RS, EO	15 patients	x	x		Improvement of PAC sensitivity based on waveform shape analysis.	x	sharpness ratio effect size: 0.86, *p* = 0.006, steepness ratio effect size: 0.68, *p* = 0.004, PAC effect size: 0.27, *p* = 0.011
Miller et al. (2019) [17].	64 channel EEG, 5 min RS, 10 s hand movements followed by 10 s rest, repeated 5 times, finger tapping on screen, 3 s movement and 7 s no movement, repeated 20 times	14 patients	x	x		Medication induced PAC attenuation corresponds to motor symptom alleviation measured using UPDRS. PAC is attenuated during voluntary movement.	x	effect size: (n/a), *p* = 0.031
Gong et al. (2020) [34].	64 channel, 5 min RS, EO	19 patients		x		Spatiotemporal characteristics of PAC.		
Gong et al. (2022) [13].	64 channel, pressing: self initiated, 3 min, 2 trials, slow tapping: 30 s, two blocks of 6 trials, fast tapping: 2 blocks; 15 s, 10 trials and 12 s, 10 trials	9 patients		x		PAC follows a movement modulation mechanism.		

Notes: ON: on-medication state, OFF, PF_OFF: unmedicated state, HC: healthy controls, EO: EEG measurements were taken with participants’ eyes open.

## Data Availability

No new data were created or analyzed in this study. Data sharing is not applicable to this article.

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
