# Peer review of "Beta–Gamma Phase-Amplitude Coupling as a Non-Invasive Biomarker for Parkinson’s Disease: Insights from Electroencephalography Studies"

_life, 2024, doi:10.3390/life14030391_

Round 1
Reviewer 1 Report
Comments and Suggestions for Authors
Dear respected Authors, I was delighted to read through your paper that I find very informative and balanced. This is the impression of a neurologist and EEG-er; as the review is addressed to a wider audience, I tried to read it from the point of a less specialised reader and find it interesting and useful again as you provide a nice introduction into the definition and significance of CFC. The more specialised part on the changes of beta-gamma coupling in PD then becomes easier to apprehend.
As a general remark on style, this article could be more succinct, avoiding redundancy and repetitions. Random example: "Tremor has been shown 232 to attenuate PAC in the STN recordings and could potentially contaminate data if present 233 during recordings [51]" - of course, if absent, tremor would not "potentially" contaminate anything - a superfluous phrase. " However, all studies excluded the possibility of the increase of 234 PAC possibly being the result of movement artifacts" - The possibility of sth possibly being so and so - tautology.
Some minor remarks. Line 88-89 - " EEG signals are proposed to be generated by excitation and inhibition of postsynaptic pyramidal neurons firing synchronously in the cerebral cortex" - I believe the standard definition is that "the electric signal recorded by EEG is generated by the ionic currents in the postsynaptic dendritic membrane of the pyramidal neurons in cortical layers IV-V" (Amzica and Lopes da Silva, 2018). "Postsynaptic pyramidal neurons"? Maybe you wanted to include in the definition of EEG the gamma activities believed to represent the spiking of neurons, if so, this should be mentioned.
Line 231 You mention that patients with "severe tremor > or = 2" were excluded. However, in the UPDRS rubric 3.15, grade 2 corresponds to "mild" tremor, 3 and 4 to "moderate" and "severe" respectively. So excluded were patients with "significant" tremor? The discussion on the possible relation of PAC changes to tremor that follows till line 251 might be better developed later when discussing the results of the studies? Here in the very beginning you are still reviewing the inclusion criteria for the 5 studies on surface EEG detected beta-gamma synchrony?
The paragraph on the possibility of PAC assessment from 2 electrode data only could be clarified. Three studies (2 of them based on the same dataset) support this possibility, yet another one (Gong et al 2020) contradicts it.
The two paragraphs 322-348 on PAC changes during walking summarise the findings of a paper that uses ECoG and is therefore not "non-invasive", not immediately within the scope of the review. While the results are interesting, in my view they pertain to the Discussion rather than to the review of non-invasive studies.
Finally Тhe Table should be corrected, the rows are not aligned in the manuscript.
In all, my minor suggestions pertain more to the style and internal logic of the narrative than to the educational value of this Review.
Author Response
Dear respected Authors, I was delighted to read through your paper that I find very informative and balanced. This is the impression of a neurologist and EEG-er; as the review is addressed to a wider audience, I tried to read it from the point of a less specialized reader and find it interesting and useful again as you provide a nice introduction into the definition and significance of CFC. The more specialised part on the changes of beta-gamma coupling in PD then becomes easier to apprehend.
Response 1.1: Dear reviewer 1, thank you very much for your thorough review and we would like to express our appreciation for the significant improvements your insights have brought to our manuscript. We are very grateful for the time and expertise you have devoted to the assessment of our paper. For your convenience, we have attached all responses and you will note that all changes have been implemented using track changes.
As a general remark on style, this article could be more succinct, avoiding redundancy and repetitions. Random example: "Tremor has been shown 232 to attenuate PAC in the STN recordings and could potentially contaminate data if present 233 during recordings [51]" - of course, if absent, tremor would not "potentially" contaminate anything - a superfluous phrase. " However, all studies excluded the possibility of the increase of 234 PAC possibly being the result of movement artifacts" - The possibility of sth possibly being so and so - tautology.
Response 1.2: We thank you for your suggestion and have worked diligently to eliminate redundancies and repetitions. If you find any other instances, please let us know and we will address them immediately.
Some minor remarks. Line 88-89 - " EEG signals are proposed to be generated by excitation and inhibition of postsynaptic pyramidal neurons firing synchronously in the cerebral cortex" - I believe the standard definition is that "the electric signal recorded by EEG is generated by the ionic currents in the postsynaptic dendritic membrane of the pyramidal neurons in cortical layers IV-V" (Amzica and Lopes da Silva, 2018). "Postsynaptic pyramidal neurons"? Maybe you wanted to include in the definition of EEG the gamma activities believed to represent the spiking of neurons, if so, this should be mentioned.
Response 1.3: This definition has been modified according to the suggestions, but the authors have chosen not to include the definition of EEG gamma activities, as this section is dedicated to the introduction to EEG, and the descriptions of the specific bands follow later. If the reviewer has any differing perspectives on this decision, please feel free to share, and we are open to further discussion.
Line 231 You mention that patients with "severe tremor > or = 2" were excluded. However, in the UPDRS rubric 3.15, grade 2 corresponds to "mild" tremor, 3 and 4 to "moderate" and "severe" respectively. So excluded were patients with "significant" tremor?
Response 1.4: Thank you for this comment, the terminology was taken from the mentioned articles, however, this comment is accurate and the terminology in the review has been changed.
The discussion on the possible relation of PAC changes to tremor that follows till line 251 might be better developed later when discussing the results of the studies? Here in the very beginning you are still reviewing the inclusion criteria for the 5 studies on surface EEG detected beta-gamma synchrony?
Response 1.5: The discussion regarding tremor and its relation to PAC has been moved to lines 314-322, however, the mention of exclusion criteria based on tremor has been left in lines 240-243, as it completes the section.
The paragraph on the possibility of PAC assessment from 2 electrode data only could be clarified. Three studies (2 of them based on the same dataset) support this possibility, yet another one (Gong et al 2020) contradicts it.
Response 1.6: The wording of this section has been improved. The studies do not provide any explanation as to why they think using data from only two electrodes could be successful or unsuccessful, and the authors have decided not to speculate.
The two paragraphs 322-348 on PAC changes during walking summarise the findings of a paper that uses ECoG and is therefore not "non-invasive", not immediately within the scope of the review. While the results are interesting, in my view they pertain to the Discussion rather than to the review of non-invasive studies.
Response 1.7: The authors find this comment fitting and have moved the section to the end of the discussion (561-588) as suggestions for future studies.
Finally Тhe Table should be corrected, the rows are not aligned in the manuscript.
Response 1.8: Thank you for this comment. We adjusted (re-centered) the text in the table.
In all, my minor suggestions pertain more to the style and internal logic of the narrative than to the educational value of this Review.
Response 1.9: The authors would like to thank reviewer 1 once again for all the valuable comments.
Reviewer 2 Report
Comments and Suggestions for Authors
The topic explored in the article named: Beta-Gamma Phase-Amplitude Coupling as a Non-Invasive Biomarker for Parkinson's Disease: Insights from Electroencephalography Studies authored by Hodnik et al. seems to be particularly intriguing. This review provides a comprehensive collection of current EEG-derived evidence regarding PAC fluctuations in Parkinson's disease (PD). It explores potential practical uses of this biomarker and puts forward recommendations for future research directions.
In general, this article is commendable. The writing and presentation adhere to scientific standards. The methodologies are suitable and in line with the stated objectives, and the conclusions align with the evidence and arguments put forth. Nonetheless, I have a few suggestions for potential enhancements
- It would be helpful if the article were improved with more studies on the relationship between EEG measures, such as beta-gamma phase-amplitude coupling, and Parkinson's disease in animal models.
- Please explain potential mechanisms underlying beta-gamma phase-amplitude coupling alterations in Parkinson's disease, as well as the significance of these findings for diagnosis, prognosis, or treatment.
- It would be useful to specify the limitations of the study.
Comments on the Quality of English Language
Minor editing of English language required.
Author Response
The topic explored in the article named: Beta-Gamma Phase-Amplitude Coupling as a Non-Invasive Biomarker for Parkinson's Disease: Insights from Electroencephalography Studies authored by Hodnik et al. seems to be particularly intriguing. This review provides a comprehensive collection of current EEG-derived evidence regarding PAC fluctuations in Parkinson's disease (PD). It explores potential practical uses of this biomarker and puts forward recommendations for future research directions.
In general, this article is commendable. The writing and presentation adhere to scientific standards. The methodologies are suitable and in line with the stated objectives, and the conclusions align with the evidence and arguments put forth. Nonetheless, I have a few suggestions for potential enhancements
- It would be helpful if the article were improved with more studies on the relationship between EEG measures, such as beta-gamma phase-amplitude coupling, and Parkinson's disease in animal models.
Response 2.1: Dear Reviewer 2, we appreciate your thoughtful comments. We have incorporated additional information in section 171-176; however, we recognize that there may be a gap in the research of animal models for investigating PAC in PD. If you have any suggestions for relevant studies that may have been overlooked, we would be grateful for your input and we will be happy to include them in our review.
- It would be useful to specify the limitations of the study.
Response 2.2: Added to 602-608. Thank you again for your time and valuable feedback.
Reviewer 3 Report
Comments and Suggestions for Authors
This paper discusses the concept of phase-amplitude coupling in EEG signals obtained from patients with Parkinson's disease. The main concept is that the coupling of beta frequencies in the phase and gamma frequencies in the amplitude can serve as a biomarker for Parkinson's Disease. The idea of phase-amplitude cross-frequency coupling in EEG signals may be clear to mathematicians, but not to neurophysiologists or clinicians. The paper briefly introduces this concept on lines 64-67: "PAC is a form of cross-frequency coupling (Figure 1), a crucial mechanism for information processing in the brain. It is involved in many higher-order cognitive processes that contribute to the successful accomplishment of everyday tasks [4]."
The table and figure legends are missing, making it it difficult to understand the content.
The authors chose five papers that investigated changes in beta-gamma phase-amplitude EEG coupling in patients with Parkinson's disease. Abbreviations "EO", "ON", "OFF", "PD_OFF", "HC" only appear in Table 1 and are not explained. This table does not appear to be informative.
The paper also mentioned other types of complex measures of EEG signals, such as cross-frequency EEG coupling and alpha-gamma phase-amplitude coupling (lines 120-170). In my opinion, this is confusing and it detracts from the main idea.
I did not find any new or original ideas in this paper. The authors mentioned some neuronal sources of beta-frequencies in EEG (basal ganglia, hippocampus). What are the neuronal mechanisms that underlie beta-gamma phase-amplitude associations, given that EEG signals are produced by the synchronization of large neuronal assemblies?
Perhaps beta-gamma phase-amplitude EEG coupling associates with a specific clinical sign of Parkinson's disease? In my opinion, this paper would benefit from a clearer characterization of the relationship between beta-gamma phase-amplitude EEG coupling and the following:
1) Medication
2) Motor disorders and other clinical signs,
3) Cognitive problems
4) Sleep disturbances.
Author Response
This paper discusses the concept of phase-amplitude coupling in EEG signals obtained from patients with Parkinson's disease. The main concept is that the coupling of beta frequencies in the phase and gamma frequencies in the amplitude can serve as a biomarker for Parkinson's Disease. The idea of phase-amplitude cross-frequency coupling in EEG signals may be clear to mathematicians, but not to neurophysiologists or clinicians. The paper briefly introduces this concept on lines 64-67: "PAC is a form of cross-frequency coupling (Figure 1), a crucial mechanism for information processing in the brain. It is involved in many higher-order cognitive processes that contribute to the successful accomplishment of everyday tasks [4]."
Response 3.1: Dear Reviewer 3, thank you for the many comments that have improved our review. The concept is explained in more detail in lines 132-139. While the explanation is limited to phase to amplitude coupling, which is a form of cross-frequency coupling, we believe that the explanation encompasses the sum of the phenomenon needed for further understanding of the concepts presented in this paper. Below is the text that was originally written, but please let us know if further explanation is needed.
[37]. Phase-to-amplitude coupling (PAC) is a form of CFC, in which a phase of a low-frequency rhythm (m) modulates the amplitude of a higher-frequency oscillation (n) [38]. In amplitude coupling the slower frequency dictates the temporal code and acts as a carrier frequency, while phase coupling improves temporal precision and is dictated by the spiking activity of the higher frequency [39]. This results in temporally coordinated neural excitability of dispersed neural populations, which in turn have increased synaptic activity and elevated impact on their select target network [40].
The table and figure legends are missing, making it it difficult to understand the content.
The authors chose five papers that investigated changes in beta-gamma phase-amplitude EEG coupling in patients with Parkinson's disease. Abbreviations "EO", "ON", "OFF", "PD_OFF", "HC" only appear in Table 1 and are not explained. This table does not appear to be informative.
Response 3.2: We thank you for these comments and have added the notes below the table. However, we are not entirely clear what you were thinking with Figure 1, as no abbreviations were used. If we have missed anything, please let us know and we will be happy to correct/add it.
The paper also mentioned other types of complex measures of EEG signals, such as cross-frequency EEG coupling and alpha-gamma phase-amplitude coupling (lines 120-170). In my opinion, this is confusing and it detracts from the main idea.
Response 3.3: We thank the reviewer for this insightful comment. While we acknowledge your perspective, we would like to offer an alternative view. The notes aim to enrich the reader's understanding of cross-frequency coupling (CFC) by providing a more comprehensive exploration of the phenomenon. Our intention is to shed light on the complicated nature of CFC and consequently provide a deeper understanding of its subset, phase-amplitude coupling (PAC). In essence, PAC can be considered as a specific manifestation of CFC, which emphasizes the importance of understanding CFC for improving our knowledge of PAC.
I did not find any new or original ideas in this paper. The authors mentioned some neuronal sources of beta-frequencies in EEG (basal ganglia, hippocampus). What are the neuronal mechanisms that underlie beta-gamma phase-amplitude associations, given that EEG signals are produced by the synchronization of large neuronal assemblies?
Perhaps beta-gamma phase-amplitude EEG coupling associates with a specific clinical sign of Parkinson's disease? In my opinion, this paper would benefit from a clearer characterization of the relationship between beta-gamma phase-amplitude EEG coupling and the following:
1) Medication
2) Motor disorders and other clinical signs,
3) Cognitive problems
4) Sleep disturbances.
Response 3.4: Thank you for these comments. Our response refers to the two paragraphs above, with some further adjustments in line with the comments of reviewer 1. With respect to the mechanisms highlighted in the comment above, we have tried to summarize them in the lines starting from 395. In relation to 1) Medication, 2) Motor disorders and other clinical signs, 3) Cognitive problems and 4) Sleep disturbances, we have included the relevant text in the following lines below. We do not currently see how this could be more clearly characterized, but are open to further suggestions.
1) Medication
Lines 69-77, 286-288, and 410-405
2) Motor disorders and other clinical signs,
Lines 224-229, and 389-401
3) Cognitive problems
Not applicable
4) Sleep disturbances.
Discussed in the introduction (from line 170) and discussion (from line 456)
With that, we would like to thank you again for your thorough review, which has improved our article.
Round 2
Reviewer 3 Report
Comments and Suggestions for Authors
Although the paper has been revised, my major concerns were not addressed.
The idea of phase-amplitude cross-frequency coupling in EEG signals may be clear to mathematicians, but not to neurophysiologists or clinicians.
Response While the explanation is limited to phase to amplitude coupling, which is a form of cross-frequency coupling, we believe that the explanation encompasses the sum of the phenomenon needed for further understanding of the concepts presented in this paper. Below is the text that was originally written, but please let us know if further explanation is needed.
Life is a multidisciplinary journal "related to fundamental themes in life sciences, from basic to applied research". To ensure effective communication, content should be presented in a manner that is easily understandable by a wide audience. The authors may refer to Figure 1 in their explanation of the phenomenon of phase-to-amplitude coupling.
The paper also mentioned other types of complex measures of EEG signals, such as cross-frequency EEG coupling and alpha-gamma phase-amplitude coupling. In my opinion, this is confusing and it detracts from the main idea.
Response 3.3: ... Our intention is to shed light on the complicated nature of CFC and consequently provide a deeper understanding of its subset, phase-amplitude coupling (PAC).
This intention seems to shift attention away from the main topic of beta-gamma phase-amplitude coupling in Parkinson's disease.
The paper is entitled "Beta-Gamma Phase-Amplitude Coupling as a Non-Invasive Biomarker for Parkinson's Disease", but this issue has been considered only in Section 2 (Current Findings Regarding PAC in PD).
The table and figure legends are missing, making it it difficult to understand the content.
Response However, we are not entirely clear what you were thinking with Figure 1,
Please read https://www.mdpi.com/journal/life/instructions.
- All Figures, Schemes and Tables should have a short explanatory title and caption.
Figure 1 lacks clarity and requires further explanation in the text.
Upon thoroughly reviewing this paper, I was unable to identify any novel or groundbreaking ideas or concepts. The authors mentioned some neuronal sources of beta-frequencies in EEG (basal ganglia, hippocampus). What are the neuronal mechanisms that underlie beta-gamma phase-amplitude associations, given that EEG signals are produced by the synchronization of large neuronal assemblies?